# Parylene-AlO$_x$ Stacks for Improved 3D Encapsulation Solutions

Sébastien Buchwalder [1,2,]*, Florian Bourgeois [3], Juan J. Diaz Leon [4], Andreas Hogg [3] and Jürgen Burger [1]

[1] School of Biomedical and Precision Engineering, University of Bern, 3010 Bern, Switzerland; juergen.burger2@unibe.ch
[2] Graduate School for Cellular and Biomedical Sciences, University of Bern, Mittelstrasse 43, 3012 Bern, Switzerland
[3] Coat-X SA, Eplatures-Grise 17, 2300 La Chaux-de-Fonds, Switzerland; bourgeois@coat-x.com (F.B.); hogg@coat-x.com (A.H.)
[4] CSEM Sustainable Energy Center, Jaquet-Droz 1, 2002 Neuchâtel, Switzerland; juan.diaz@csem.ch
* Correspondence: sebastien.buchwalder@unibe.ch

**Abstract:** The demand for ultra-tight encapsulation solutions with excellent barrier and high conformality properties has increased in recent years. To meet these challenges, thin-film barrier coatings have emerged as a promising solution. In this study, we investigate well-established silicon-based plasma-enhanced chemical vapor deposition (PECVD) and metal oxide atomic layer deposition (ALD) barrier coatings deposited at low temperatures ($\leq$100 °C) regarding their abilities to address high-level 3D encapsulation applications. Various combinations of such layers are evaluated by measuring the water vapor transmission rate (WVTR) and considering the conformality properties. The impact and the benefits of the organic film integration, namely parylene VT4 grade, on the barrier performances is assessed. Among these combinations, parylene-AlO$_x$ stack emerges as one of the most effective solutions, obtaining a WVTR of $3.1 \times 10^{-4}$ g m$^{-2}$ day$^{-1}$ at 38 °C and 90% relative humidity conditions.

**Keywords:** barrier layer; thin film; parylene; atomic layer deposition; water vapor transmission rate; conformality



## 1. Introduction

In recent years, there has been growing demand for high-performance encapsulation solutions that can provide excellent barrier properties while offering interesting mechanical properties. Conventional encapsulation solutions, such as metal or glass casings, provide reliable protection but suffer from certain limitations regarding the potential for the miniaturization and flexibility of the protective case. One promising approach to address this challenge is the use of thin-film barrier coatings. These coatings, typically made of alternating organic–inorganic layers, can provide enhanced protection by combining the unique properties of both types of materials. Parylenes, or poly-*p*-xylylene and its derivatives, are polymeric films deposited at ambient temperature using the Gorham process through chemical vapor deposition [1]. Due to parylene properties and by combining it with inorganic layers, it is possible to create a conformal, pinhole-free barrier layer with exceptional barrier properties, even at a low deposition temperature. Water vapor transmission rate (WVTR) measurement is an essential technique employed in various industries to assess the barrier properties of materials. WVTR is expressed in units of grams per square meter per day (g m$^{-2}$ day$^{-1}$) and represents the quantity of moisture passing through the test specimen over a time period and area. Currently, a range of instruments exist to quantify WVTR, involving various technologies, such as gravimetric evaluation [2], coulometric testing [3], calcium corrosion [4] and tunable diode laser absorption spectroscopy [5], which stands out as one of the most sensitive methods, allowing the measurement of WVTR in the extremely low range of $10^{-6}$ g m$^{-2}$ day$^{-1}$.

Micro- and optoelectronics are commonly cited as applications requiring highly conformal thin-film encapsulation solutions. Integrated circuits (ICs) and microelectromechanical systems (MEMS) devices require encapsulation to protect the sensitive components from environmental factors such as moisture, oxygen and chemical contaminants, while optoelectronic devices such as photovoltaic modules, display panels and organic electronic technologies, including organic light-emitting diodes (OLEDs), need ultra-tight, in the range of $10^{-6}$ g m$^{-2}$ day$^{-1}$, transparent and flexible barrier alternatives [6–8]. In addition, thermally sensitive bioelectronic and biosensor applications demand flexible protection compatible with low-temperature processes [9–11]. Energy storage and conversion such as batteries, fuel cells and other energy storage and conversion devices require efficient thin-film encapsulation to prevent moisture and oxygen penetration into the active materials, which can degrade their performance and reduce their lifespan [12–16]. Cros et al. [17] demonstrated that, for instance, organic solar cells can be protected with medium barrier materials with a WVTR of around $10^{-3}$ g m$^{-2}$ day$^{-1}$. Advanced developments in the biomedical domain involve a barrier coating solution that prevents the interactions with surrounding tissue, ensuring long-term biocompatibility and enabling the further miniaturization of implantable devices [18–21].

In this study, we systematically investigated the barrier performances of inorganic coatings deposited via plasma-enhanced chemical vapor deposition (PECVD) and atomic layer deposition (ALD). Silicon-based layers and metal oxides were characterized by measuring their WVTR, employing two techniques. For WVTR values up to $10^{-3}$ g m$^{-2}$ day$^{-1}$, an electrolytic detection sensor based on a coulometric method was applied, following the international standard ISO 15106-03 [3]. For lower WVTR values, the diode laser spectroscopy method was employed; this specific equipment was designed to determine WVTR for materials with ultra-high barrier properties, enabling the detection of values down to $10^{-6}$ g m$^{-2}$ day$^{-1}$. Organic film, fluorinated parylene VT4 grade, deposited via a low-pressure chemical vapor deposition (LPCVD) process, was then combined with inorganic layers to build up organic–inorganic stacks, and the benefits in terms of barrier, mechanical and chemical properties were evaluated. In addition, to assess the conformality, micro-channel structures on the silicon wafer were manufactured in order to determine the aspect ratio (AR) of the coatings investigated in this study.

## 2. Materials and Methods

### 2.1. Substrate

For the evaluation of the barrier properties, the barrier layers were deposited on two different substates: polyethylene terephthalate (PET) Melinex Peelable-Clean-Surface (PCS) grade and polyimide (PI) Kapton HN grade substrate.

### 2.1.1. PET Melinex PCS Grade Substrate

Firstly, the ultra-clean PET Melinex PCS grade substrate, 125 μm thick, produced by DuPont Teijin Films™ [22], was employed to measure the WVTR of the barrier layers. A liner film, protecting the PET substrate, was removed before the deposition in a clean environment to avoid as much particle contamination as possible. The PET Melinex PCS substrate, presenting a melting point between 255 °C and 260 °C, exhibited a WVTR of 6 g m$^{-2}$ day$^{-1}$ at 38 °C and 90% relative humidity (RH).

### 2.1.2. PI Kapton HN Grade Substrate

Next, 125 μm thick PI Kapton HN grade substrate, one of the most thermally stable polymers with a decomposition point around 500 °C [23], was used to evaluate the barrier coating on common substrate. Kapton HN type is the recommended choice for applications that require a film with an excellent balance of properties over a wide range of temperatures. The PI film shows a WVTR of 15 g m$^{-2}$ day$^{-1}$ at 38 °C and 90% RH. The substrate was cleaned with isopropanol solution and dried with a high-purity nitrogen blow gun before the deposition processes.

### 2.2. Inorganic Layer Deposition

#### 2.2.1. PECVD Silicon Oxide, SiO$_x$

Silicon oxide, SiO$_x$, was deposited via a capacitively coupled plasma (CCP) deposition using hexamethyldisiloxane (HMDSO) as a silicon-content precursor and oxygen (O$_2$). CCP deposition was performed in a CX-30 PC deposition system, provided by Coat-X SA (La Chaux-de-Fonds, Switzerland). Amorphous SiO$_x$ was deposited with a flow rate ratio of 1:10 at a pressure of 36 μbar with a RF power of 50 W applied at 13.56 MHz. The deposition temperature was maintained at around 30 °C, and the layer thickness was set to 160 ± 10 nm to avoid excessive internal stress in the layer.

#### 2.2.2. PECVD Silicon Nitride, SiN$_x$

Amorphous silicon nitride, SiN$_x$ was deposited with silane (SiH$_4$) precursor combined with nitrogen (N$_2$) and ammonia (NH$_3$). Inductively coupled plasma (ICP) deposition was performed in an Oxford Plasma 80 PECVD system, provided by Oxford Instruments GmbH (Abingdon, UK). The flow rate ratio was 1:50:1 (SiH$_4$; N$_2$; NH$_3$), and the deposition pressure was 0.87 mbar. The depositions were achieved at 20 W RF power with a temperature fixed to 100 °C. The layer thickness was again targeted to 160 ± 10 nm.

#### 2.2.3. ALD Metal Oxides

ALD metal oxides were deposited in a FlexAl PE-ALD system in thermal mode provided by Oxford Instruments GmbH. Fully amorphous aluminum oxide (AlO$_x$), titanium oxide (TiO$_x$) and hafnium oxide (HfO$_x$) were deposited using, trimethylaluminum (TMA) and tetrakis(dimethylamino)titanium (TDMAT) and tetrakis(dimethylamido)hafnium (TDMAH), respectively, as metalorganic precursors. The deposition temperature was set at 100 °C, and the thickness of the layers was set to 45 nm ± 5 nm.

### 2.3. Organic Layer Deposition

Poly(tetrafluoro-*p*-xylylene), commonly identified as fluorinated parylene VT4, was deposited at room temperature using the LPCVD technique based on the Gorham route [1]. The parylene deposition process contained three stages: sublimating the solid dimer into vapor, disassociating the dimers into monomers and, lastly, condensing the monomers within the chamber to yield a polymeric film. To ensure a constant pressure of 80 μbar in the deposition chamber, the sublimation temperature was carefully regulated from 80 to 150 °C. The pyrolysis temperature was set at 700 °C to cleave the dimers into monomers. Finally, the monomers underwent condensation and polymerization within the chamber to form a polymeric film at room temperature. The parylene film structure was considered to be a semi-crystalline material showing both crystalline domains and amorphous regions [24,25]. Depending on process parameter, the average molecular weight of the parylene chain was $2.5 \times 10^5$–$4 \times 10^5$ g mol$^{-1}$, assuming that the chain ends were uniformly distributed throughout the film [26]. Parylene VT4 grade, known to possess a superior thermal stability and a low dielectric constant with values in the range of 2.05–2.35. [27]. As previous studies have shown, parylene VT4 has a melting point above 400 °C, but the polymer begins to degrade above 300 °C [24,28]. The decomposition temperature was found to be between 495 °C and 510 °C [24,27]. Compared to parylene N grade, the basic parylene form consisting of a linear carbon–hydrogen molecule structure, parylene VT4 grade incorporates fluorine atoms in the aromatic sites. The chemical structures of parylene N and parylene VT4 grades are illustrated in Figure 1a,b, respectively.

Parylene film was deposited in a CX-30 PC hydride machine, able to deposit inorganic layers via CCP, provided by Coat-X SA. Before the deposition of a 2 ± 0.2 μm thick parylene film, an adhesion promoter, namely methacryloxypropyl trimethoxysilane (A-174), was evaporated for 3 min in the chamber.

**Figure 1.** Chemical structure of (**a**) parylene N grade, the basic form of parylene, consisting of a linear carbon–hydrogen molecule structure, and (**b**) parylene VT4 grade incorporating fluorine atoms in the aromatic sites.

### 2.4. Characterization Method

The water vapor transmission rate (WVTR) was determined using two different methods but applying the same measurement conditions: a temperature of 38 °C and a relative humidity of 90%. The measurements, in which each layer combination was measured at least three times ($n \geq 3$), were performed by Coat-X SA in collaboration with the Fraunhofer Institute for Organic Electronics, Electron Beam and Plasma Technology (FEP), Dresden. In addition to WVTR measurements, conformality investigations were conducted to evaluate the ability of parylene and inorganic layers to fill in cavities.

### 2.4.1. Conformality Investigation

To assess the conformality of the coatings investigated in this study, micro-channel structures on silicon wafer were manufactured using the standard lithography process. These micro-channels, measuring 50 μm in depth and width, were etched onto the silicon wafer. Subsequently, a covering glass slide was placed atop the wafer during the deposition process to enclose the channels, resulting in the formation of micro-channels. An open cavity positioned at the termination of the channel ensured channel entrances. Figure 2 illustrates the micro-channel structure on silicon wafer, both seen from above (Figure 2a) and with a lateral cut (Figure 2b). Additional materials can be found in the Appendix A.

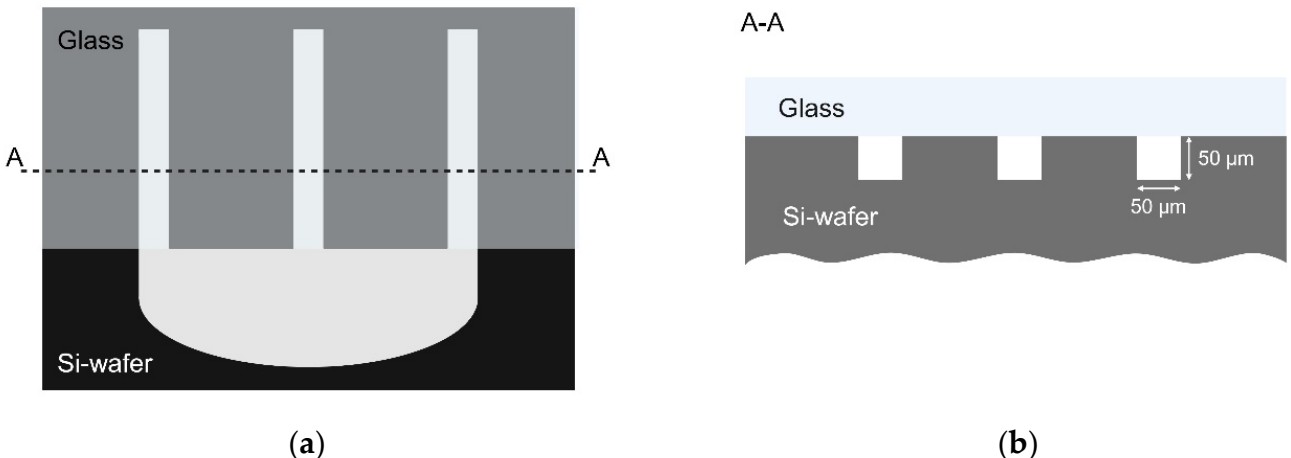

**Figure 2.** Illustrations of the micro-channel structures on silicon wafer for the conformality investigations: (**a**) top view of the structure comprising the micro-channel and the cavity in front of them; (**b**) lateral cut view of the 50 μm channels etched in a silicon wafer and covered with a glass slide.

By examining these structures, we derived a lateral aspect ratio for the coatings. To quantify this aspect ratio, the penetration length of the coating was measured. This measurement then allowed us to calculate the aspect ratio (AR) using the following formula:

$$\text{AR} = \frac{\text{penetration length}}{\text{channel width}}, \tag{1}$$

### 2.4.2. WVTR Electrolytic Detection Method

WVTR measurements were conducted, in accordance with international standard ISO 15106-03, using a water vapor permeability (Wasserdampf-Durchlässigkeits, WDDG, Munich, Germany) instrument containing an electrolytic detection sensor provided by Brugger Feinmechanik GmbH (Munich, Germany). The sample, with a diameter of 15 cm, was placed in a test cell consisting of a dry chamber and a controlled humidity chamber. The controlled humidity chamber maintained a constant water vapor pressure using a sulfuric acid solution. The sample, with the coated side facing the dry chamber, was exposed to a dry nitrogen carrier gas. The water vapor permeating through the sample was transported by the carrier gas to the electrolytic cell. By applying a DC voltage, the water vapor in the carrier gas was electrolytically decomposed into hydrogen and oxygen. The mass of the permeating moisture per time interval was determined by analyzing the electrolytic current and dividing it by the area of the test specimen. The water vapor transmission rate was calculated using the following equation:

$$\text{WVTR} = \frac{I}{A} 8.067, \tag{2}$$

where WVTR (g m$^{-2}$ day$^{-1}$) is the water vapor transmission rate of the specimen, expressed in grams per square meter and per day; $A$ (m$^2$) is the transmission area of the test specimen in square meter; $I$ (A) is the electrolytic current in amperes; and 8.076 is the instrument constant.

### 2.4.3. WVTR Diode Laser Spectroscopy Method

For lower WVTR values, diode laser spectroscopy method was employed. The Hi-BarSens equipment, provided by the company Sempa GmbH (Dresden, Germany), was developed for the determination of water vapor permeation rates of an ultra-high barrier material, which allowed detection until $10^{-6}$ g m$^{-2}$ day$^{-1}$. The 20 cm diameter sample divided the system into two parts. On the upper side, a defined water vapor concentration was generated at a set temperature. The measurement chamber where a tunable diode laser could quantitatively determine the permeated moisture concentration was located below the sample. The detection method was single-line absorption spectroscopy. The variation in the intensity due to absorption of the water vapor molecules between the transmitter and receiver gave direct and absolute measurements for water vapor permeation rate. The water vapor transmission rate was calculated using the following equation:

$$\text{WVTR} = \frac{c\,Q_{N2}\,P\,M_{H2O}}{A\,R\,T\,(1 - c)}, \tag{3}$$

where WVTR (g m$^{-2}$ day$^{-1}$) is the water vapor transmission rate of the specimen, expressed in grams per square meter and per day; $c$ (ppm) is the concentration of water; $Q_{N2}$ (sccm) is carrier gas flow rate; $P$ (Pa) is the pressure; $M_{H2O}$ (g mol$^{-1}$) is molar mass of water molecule; $A$ (m$^2$) is the area of the test sample; $R$ (J mol$^{-1}$ K$^{-1}$) is the gas constant; and $T$ (K) is the measurement temperature.

## 3. Results and Discussion

### 3.1. Conformality

Figure 3 illustrates the penetration ability of PECVD SiO$_x$ (Figure 3a), ALD AlO$_x$ (Figure 3b) and LPCVD parylene VT4 (Figure 3c) layers in micro-channels, measuring 50 μm in depth and width. Inorganic PECVD and ALD layers were deposited at 160 nm and 45 nm, while the parylene film was made to be 605 nm thick to determine the optical interferences.

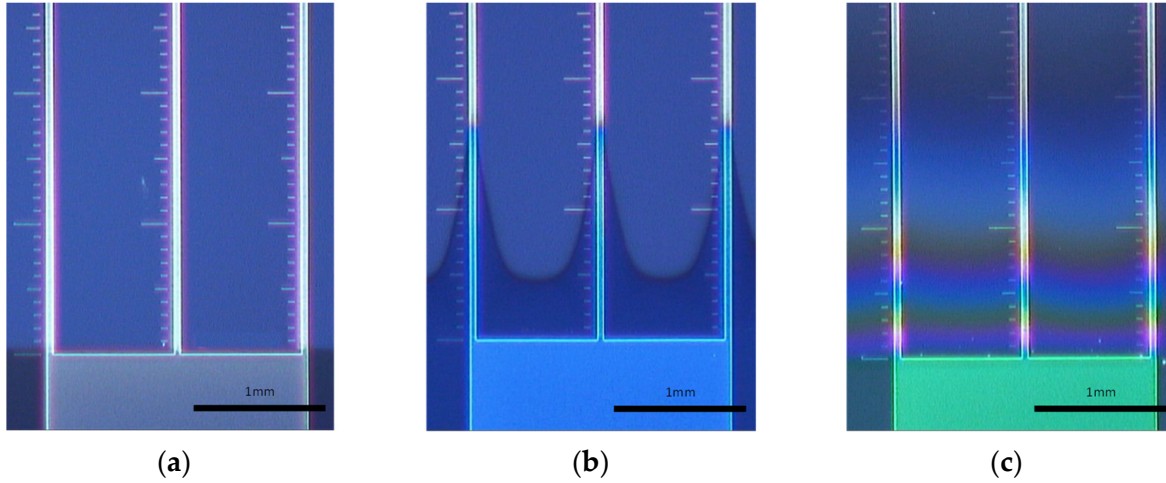

**(a)**　　　　　　　　　　　**(b)**　　　　　　　　　　　**(c)**

**Figure 3.** Silicon wafer micro-channel structures after deposition: (**a**) SiO$_x$ (160 nm) deposited via PECVD; (**b**) AlO$_x$ (45 nm) deposited via ALD; (**c**) parylene film (605 nm) deposited via LPCVD.

Optical microscopy imaging allowed us to measure the penetration length and hence determine the lateral aspect ratio (AR) of each coating. The SiO$_x$ layer deposited via PECVD exhibits a penetration length of about 0.08 mm, corresponding to an AR of 1.6:1. On the other hand, the AlO$_x$ layer grown using ALD demonstrates a penetration length of 1.6 mm, resulting in an AR of 32:1. Similarly, the parylene film displays a penetration length of approximately 1.7 mm, yielding an AR of 34:1.

Table 1 presents a summary of the measured penetration abilities, alongside their corresponding ARs. Additionally, comparative AR values are provided from the existing literature. An important contrast emerges between PECVD SiO$_x$ and ALD AlO$_x$ layers, as the ALD technique shows a penetration ability that is 20 times higher than that of the PECVD technique. The existing literature supports the notion that PECVD layers typically exhibit an AR approaching a maximum of 2:1 [29,30]. In contrast, ALD processes are known for yielding higher ARs compared to PECVD, a characteristic also confirmed via our measurements. For instance, Gabriel et al. [31] documented an AR of 38:1 for AlO$_x$ deposited using ALD techniques, while a higher equivalent AR can be achieved if a higher process temperature is used [32,33]. Kim et al. [34] indicated that thermal ALD oxide layers deposited at 200 °C can display an AR of 200:1. Parylene films, in particular parylene C grade, have been investigated for applications involving high ARs [35–37]. Suzuki et al. [38] indicated that an AR ranging from 10:1 to 20:1 was achievable with parylene C. Confirming parylene's VT4 conformal properties is difficult since this grade has been synthesized quite recently compared to the other types, and related studies are rare. However, a fluorinated parylene variant, known as parylene AF4, has exhibited exceptional promise, with an equivalent AR higher than 78:1 [39]. This variant is known for its superior penetration abilities, outperforming other parylene types in this regard.

**Table 1.** Summary of the penetration length measured for each coating and the corresponding AR. Equivalent AR values provided by the literature are also presented.

| Coating | Penetration Length [mm] | AR in this Study | AR in the Literature |
|---|---|---|---|
| PECVD SiO$_x$ | 0.08 | 1.6:1 | 2:1 [29,30] |
| ALD AlO$_x$ | 1.6 | 32:1 | 38:1 [31] |
| LPCVD Parylene VT4 | 1.7 | 34:1 | - |

### 3.2. PECVD SiO$_x$ and SiN$_x$ Barrier Layers

Figure 4 provides a schematic representation of the different layer combinations evaluated for WVTR measurements. PECVD silicon-based inorganic layers exhibited a thickness of $160 \pm 10$ nm, while the parylene films displayed a thickness of $2 \pm 0.2$ μm, deposited on PET Melinex PCS substrate, 125 μm thick.

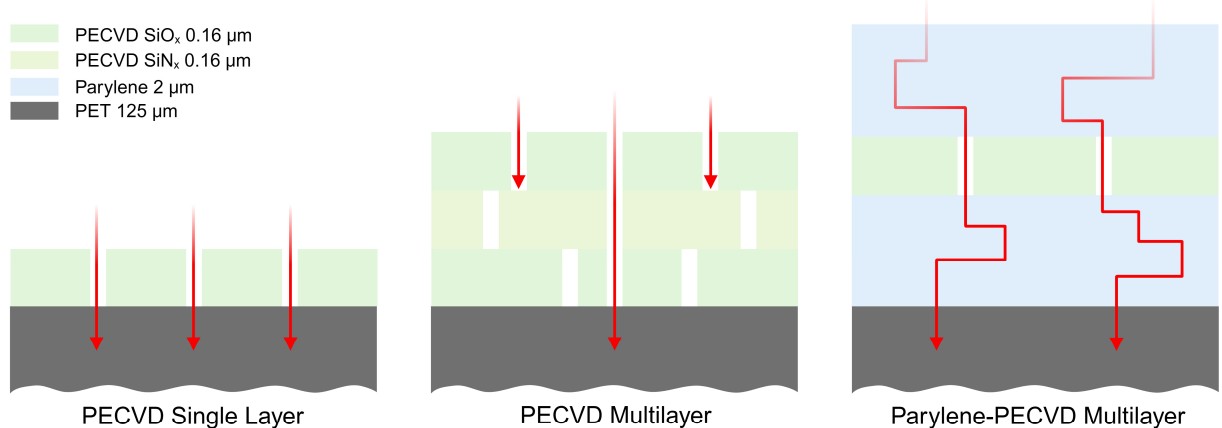

PECVD SiO$_x$ 0.16 μm
PECVD SiN$_x$ 0.16 μm
Parylene 2 μm
PET 125 μm

PECVD Single Layer — PECVD Multilayer — Parylene-PECVD Multilayer

**Figure 4.** Three combinations investigated regarding WVTR measurements: PECVD single layer, SiOx and SiN$_x$ as single layers, PECVD multilayer, combinations of SiO$_x$ and SiN$_x$ layers and a Parylene-PECVD multilayer where the inorganic layer is encapsulated by parylene films.

Figure 5 presents WVTR values of SiO$_x$ and SiN$_x$, abbreviated as SO and SN, respectively, of the combination illustrated in Figure 4. WVTR values for SiO$_x$ and SiN$_x$, as monolayers, are limited to $1.8 \times 10^{-2}$ g m$^{-2}$ day$^{-1}$ and $4 \times 10^{-3}$ g m$^{-2}$ day$^{-1}$, respectively. To increase the barrier performances, the combination of PECVD layers is necessary. The bilayer combination SiO$_x$/SiN$_x$ greatly decreases the WVTR value, reaching $9 \times 10^{-4}$ g m$^{-2}$ day$^{-1}$, while an additional layer of SiO$_x$ does not improve the barrier properties, obtaining a WVTR of $1.5 \times 10^{-3}$ g m$^{-2}$ day$^{-1}$. Finally, the encapsulation in a parylene (Px) "sandwich" of SiO$_x$ and SiN$_x$ monolayers yields WVTR values of $7 \times 10^{-3}$ g m$^{-2}$ day$^{-1}$ and $2 \times 10^{-3}$ g m$^{-2}$ day$^{-1}$, respectively.

Our results have shown that the combination of two or more PECVD layers greatly improves the barrier performances compared to a single layer. These results can be explained by the fact that water diffusion through PECVD layers is mainly governed by the presence of pinhole defects [40–42]. In consequence, the superposition of PECVD layers decouples the defects existing in the previous layer and hence significantly enhances barrier properties by blocking the diffusion through preceding layer defects. The main advantages of using PECVD layer combinations for barrier solutions are the high deposition rate (i.e., about 20 nm/min) combined with a large-scale deposition capability and a low deposition temperature compatibility compared to the common CVD process. However, SiO$_x$ and SiN$_x$ PECVD layers have limited conformality due to the directionality of the deposition technique [43]; conformality investigations, as presented above, and previous studies have indicated that similar layers exhibit an aspect ratio of approximately 2:1 [29,30]. Furthermore, the addition of PECVD layers to form an inorganic multilayer solution increases

the internal stress formation in the stack, which can lead to failures such as warping, delamination and even cracks [44]. This reason could explain why the trilayer SO/SN/SO does not show better barrier properties than those of the bilayer SO/SN. In addition, the superposition of inorganic layers reduces the flexible and stretchable properties of the barrier stack [45]. To avoid these limitations, a polymeric film made of fluorinated parylene VT4 was integrated to build up organic–inorganic combinations. Parylene encapsulation has a favorable effect on $SiO_x$ and $SiN_x$ single layers, having a WVTR improvement factor of 2.6 and 2, respectively, even if the parylene VT4 itself exhibits poor WVTR behavior, i.e., $72 \text{ g m}^{-2} \text{ day}^{-1}$ for a 4 µm thick film [24]. The parylene film on top of the stack also has the advantage of increasing the mechanical robustness and hence avoiding the scratch deterioration of the PECVD layers through handling or wear and tear. Finally, the first parylene film acts as a stress-releasing layer, which helps to avoid defect site formation and hence permits the buildup of robust a parylene-PECVD multilayer stack [19].

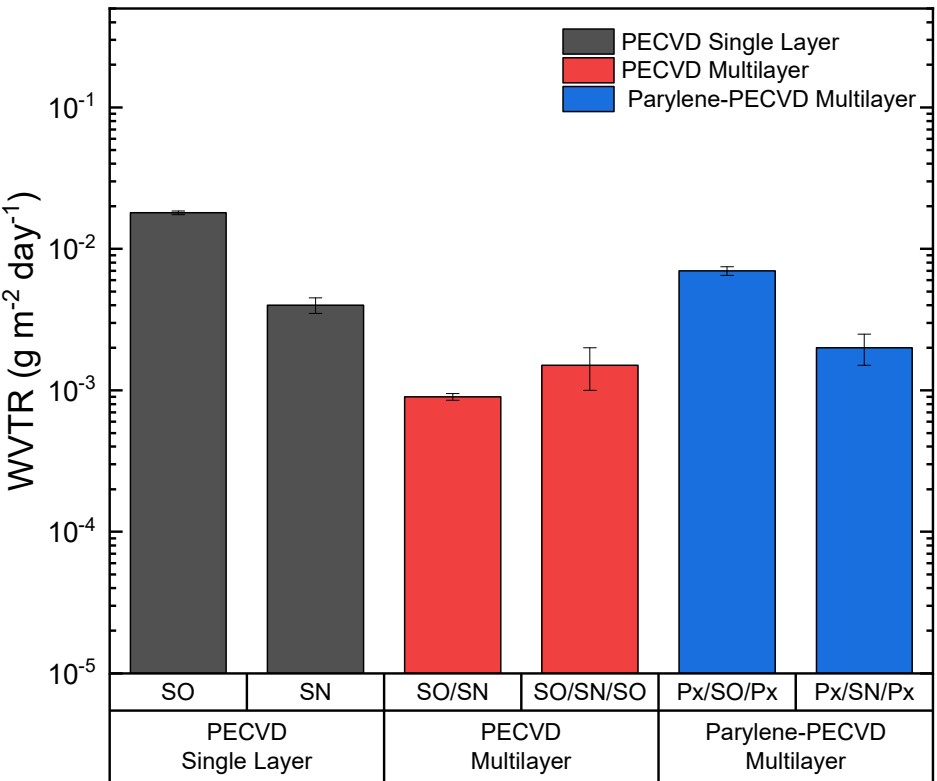

**Figure 5.** WVTR values of PECVD $SiO_x$ and $SiN_x$, abbreviated as SO and SN, respectively, as a single layer, in combination with a PECVD inorganic multilayer and fully encapsulated by parylene films (Px) as an organic–inorganic multilayer.

### 3.3. ALD Metal Oxide Barrier Layers

3.3.1. WVTR Results on PET Melinex PCS Substrate

Figure 6 illustrates the combination categories, containing ALD metal oxides and PECVD silicon oxide in association with parylene films, as assessed through WVTR measurements. Silicon-based and parylene layers exhibited a similar thickness, as described in Section 3.1, while ALD metal oxides demonstrated a thickness of $45 \pm 5$ nm. All the coatings were again deposited onto an ultra-clean PET Melinex PCS substrate with a thickness of 125 µm.

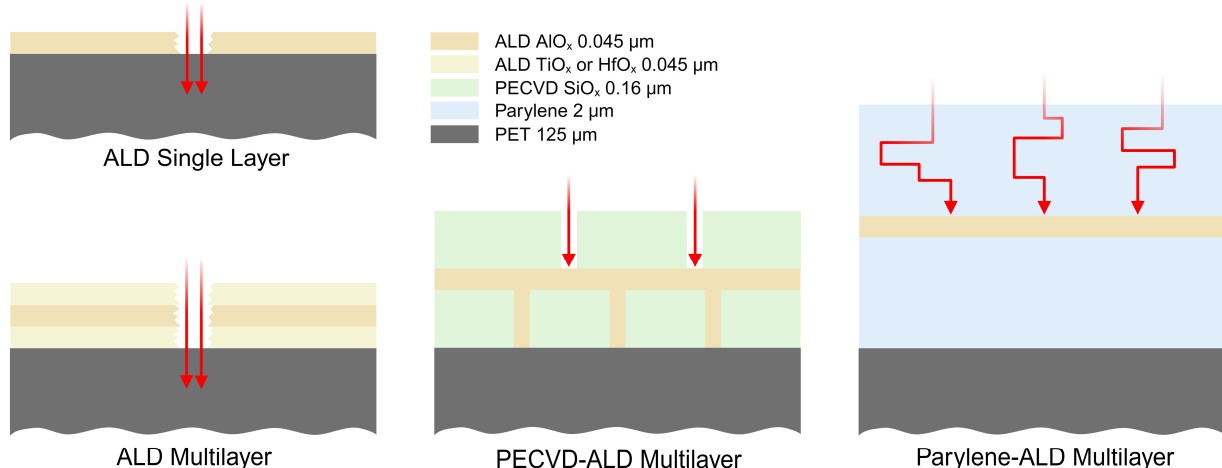

**Figure 6.** Combination configurations studied regarding their WVTR measurements. Fully ALD-based solutions, single layer and multi-layer groups, combinations of PECVD and ALD multilayer structures and organic–inorganic multilayer-associated parylene films and ALD metal oxides.

Figure 7 shows the WVTR values of aluminum oxide, $AlO_x$ (A), deposited via thermal ALD as a monolayer, combined with other oxides and parylene. The ALD Multilayer category combines $AlO_x$ with other ALD metal oxides, such as titanium oxide (T) and hafnium oxide (H), while the PECVD Multilayer group combines $SiO_x$ (SO) and $SiN_x$ (SN) with $AlO_x$. The $AlO_x$ layer directly deposited onto the PET Melinex PCS substrate yields a limited WVTR of $2.2 \times 10^{-2}$ g m$^{-2}$ day$^{-1}$, while the encapsulation of $AlO_x$ with other ALD oxides is not sufficient to improve the WVTR. In contrast, the combination with silicon oxide layers appears to be highly beneficial. Indeed, the inorganic trilayer SO/A/SO demonstrates an excellent WVTR value of $2.4 \times 10^{-4}$ g m$^{-2}$ day$^{-1}$. Finally, the last group evaluates the integration of parylene films in combination with $AlO_x$ to form bilayers and a fully parylene encapsulated structure. The results indicate that the integration of parylene film greatly improves the WVTR value for the bilayer A/Px and trilayer Px/A/Px, indicating a WVTR $3.1 \times 10^{-4}$ g m$^{-2}$ day$^{-1}$ and WVTR of $4.3 \times 10^{-4}$ g m$^{-2}$ day$^{-1}$, respectively.

ALD oxides, as single layers or multilayers directly deposited onto the substrate, reveal barrier performance limitations. Indeed, $AlO_x$ alone or encapsulated by $TiO_x$ or $HfO_x$ do not exhibit WVTR values under $10^{-2}$ g m$^{-2}$ day$^{-1}$. In contrast, the PECVD-ALD combinations show important improvement compared to the respective single layers. Two hypotheses can explain those results: Firstly, ALD single layers and multilayers have poor WVTR due to their limited robustness. Indeed, ALD layers with a thickness less than 50 nm can be damaged or degraded through handling or during measurements. The second hypothesis states that the permeability of PECVD layers is limited by defects, and, thus, the combination with $AlO_x$ allows it to fill in pinholes and/or nanocrack defects due to its important conformality and high-aspect-ratio properties [33,34,46]. As shown, the PECVD-ALD combination greatly improves the barrier performance; nevertheless, this solution introduces limitations regarding the conformality due to the directionality of the PECVD process. To overcome this limitation, $AlO_x$ is combined with parylene VT4 film, which is a highly conformal, uniform and pinhole-free coating [47]. The bilayer Px/A shows that using a parylene interlayer between the substrate and the $AlO_x$ does not increase the barrier performance. In this case, PET Melinex PCS substate can be considered as an ideal substrate due to its highly clean and particle-free surface. On the other hand, the bilayer A/Px demonstrates a great WVTR improvement and validates the requirement of $AlO_x$ to be capped by an additional layer to prevent its chemical and/or mechanical degradation. Parylene film can act as a mechanical protection against scratches and the deterioration of the ALD layers, which can occur during the handling or measurement steps. Furthermore, parylene, known for having excellent ionic barrier properties, can be considered as a chemical protective coating against $AlO_x$ hydrolysis [48]. Indeed, $AlO_x$

deposited via ALD can manifest chemical instability and be prone to easy dissolution through hydrolysis when directly exposed to aqueous solutions [49–52]. The results of the bilayer A/Px and the trilayer Px/A/Px tend to confirm that the capping parylene layer is necessary to maintain and preserve high barrier performance given by the $AlO_x$ layer.

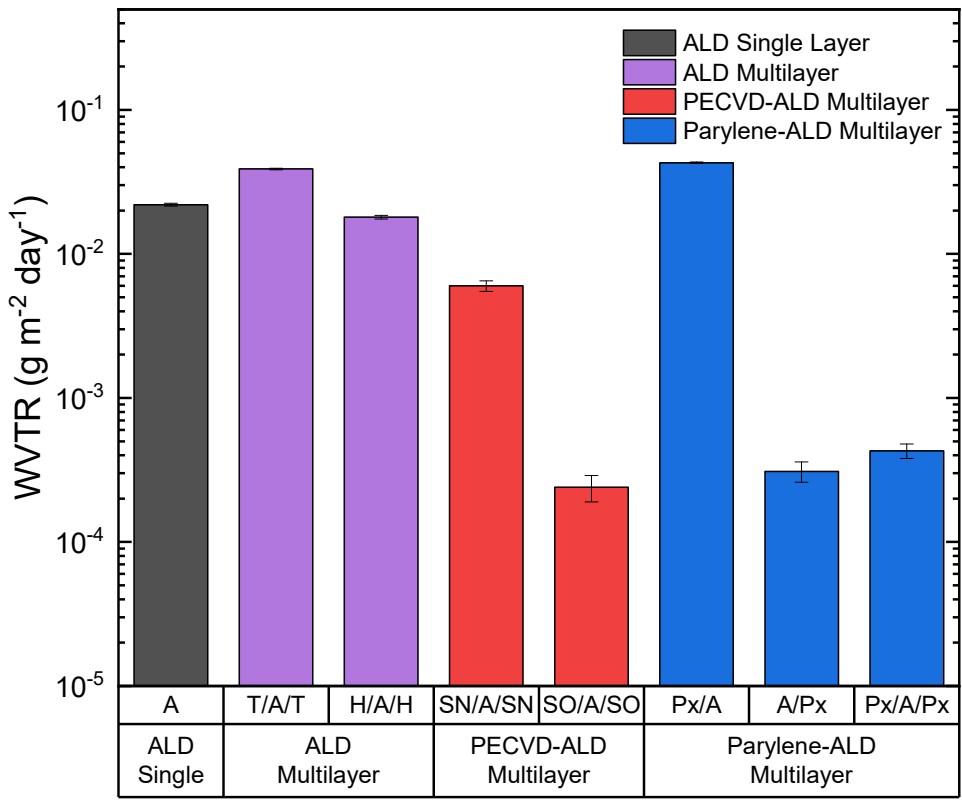

**Figure 7.** WVTR values of ALD aluminum oxide, $AlO_x$ (A), as single layer, in combination with others ALD and PECVD oxides and, finally, with parylene films (Px). The highly conformal and uniform bilayer A/Px exhibits a WVTR $3.1 \times 10^{-4}$ g m$^{-2}$ day$^{-1}$, which is extremely attractive for complex 3D geometry encapsulation.

3.3.2. WVTR Results on PI Kapton HN Substrate

Similar to the results shown above, Figure 8 displays, on the hatched columns, the WVTR results of $AlO_x$ on the PI Kapton HN substrate in comparison to the previously presented results on the PET Melinex PCS substrate. The two combinations on the PI substrate, namely the bilayer $AlO_x$/parylene and the trilayer parylene/$AlO_x$/parylene, exhibit WVTRs of $1.2 \times 10^{-2}$ g m$^{-2}$ day$^{-1}$ and $7.6 \times 10^{-4}$ g m$^{-2}$ day$^{-1}$, respectively.

The blue light columns indicate that only adding a capping layer is no longer enough to achieve a low WVTR on a PI substrate, which can be considered as common and non-ideal compared to the ultra-clean PET Melinex PCS substrate. In this case, $AlO_x$ has to be fully encapsulated by parylene film in order to reach a comparative WVTR value, as exhibited by the bilayer $AlO_x$/Parylene on the PET substrate. On the PI substrate, the first parylene layer can be considered as an interlayer between the substrate and the $AlO_x$, which smooths the asperities and encapsules the possible particles present on the substrate. Therefore, incorporating a first layer of parylene provides a way to regulate surface properties prior to ALD deposition and potentially offers a protection solution that is not dependent on the substrate surface condition, thus enabling its application on diverse materials. Finally, as mentioned before, the capping parylene layer can be considered to be an essential mechanical and chemical protective layer, as it is necessary to preserve the barrier properties of $AlO_x$.

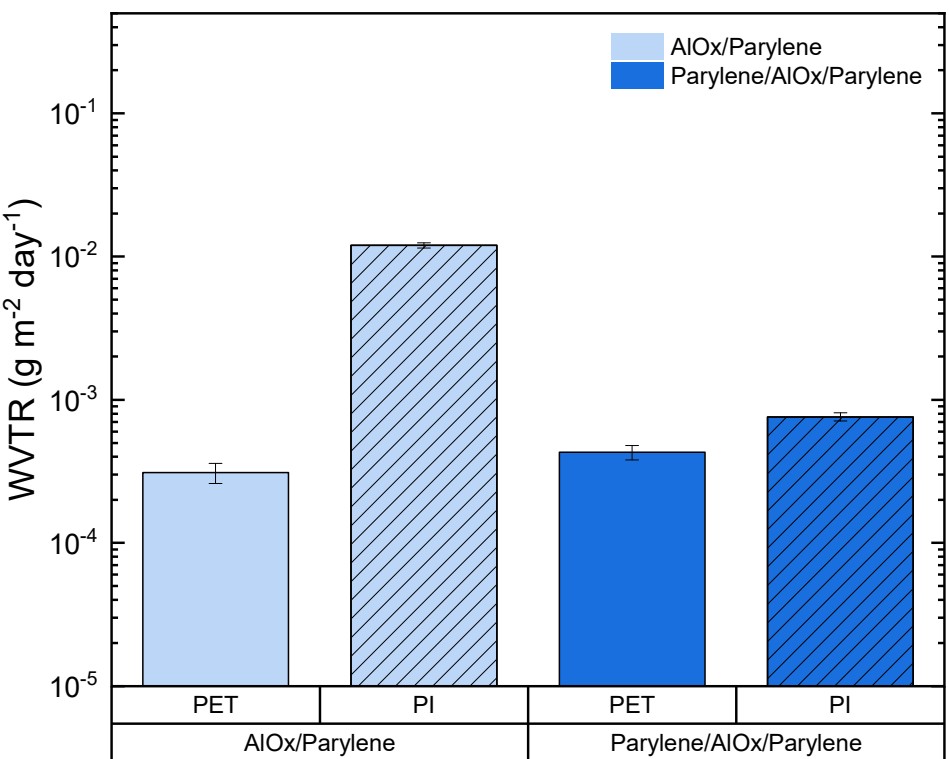

**Figure 8.** Parylene-AlO$_x$ combination comparisons on two different substrates: clean polyethylene terephthalate (PET) Melinex PCS grade and common polyimide (PI) Kapton membrane. The ALD AlO$_x$ barrier layer has to be fully encapsulated by parylene films on a non-ideal substrate (PI) to reach a WVTR value in the range of $10^{-4}$ g m$^{-2}$ day$^{-1}$, being an encapsulation solution that is potentially not dependent on the substrate's surface condition.

As demonstrated by the results, the combination of ALD with parylene offers several benefits. Firstly, it can significantly enhance the barrier properties of the ALD layer, thereby improving its overall performance as a protective coating. Additionally, the use of parylene with ALD results in excellent conformality, as well as high flexibility and stretchability, making it suitable for applications in which the coated substrate is subject to mechanical stress or deformation. Moreover, the obtained WVTR values, measured over a large area (approximately 150 cm$^2$), provide a comprehensive assessment of the global barrier performance, in contrast to intrinsic WVTR values, which exclude the pinhole or defect contributions, as often stated in the literature. Furthermore, the use of this combined technique can provide a solution that is effective on common substrates, which are typically more challenging to coat due to their variable surface properties. Overall, the use of ALD barrier layers with parylene represents a promising approach for achieving enhanced barrier properties, improved conformality and increased flexibility, all while being applicable to a wider range of substrates. This solution is also compatible with the superposition of parylene-AlO$_x$ layers, building up multiple dyad structure, thereby achieving higher barrier performance. For further flexibility-related studies, optimizing the number of layers in a multilayer, as well as the thickness of both organic and inorganic layers, while maintaining excellent barrier properties, would be of great interest.

To complete the discussion, the following paragraph provides some considerations regarding the effect of temperature on ALD depositions in term of layer properties. When comparing depositions conducted at 100 °C to those performed at lower temperatures, several important insights can emerge. Firstly, lower temperatures can lead to the decreased mobility of precursor molecules and affect their distribution during the deposition process, reducing film uniformity and conformality. Consequently, achieving high uniformity and conformality becomes more challenging. Moreover, a reduced temperature necessitates an

extension of the process' duration due to the need for longer exposure and purge times. Those adjustments are essential to prevent the overlapping of the exposure and purge steps, thereby avoiding a continuous deposition mode, similar to common CVD processes. Low deposition temperatures can give rise to inefficiencies in the purging process, decelerating the desorption rate of precursors from the surfaces [53]. This effect is particularly pronounced when considering $TiO_x$ layers, deposited via the TDMAT precursor, where the process duration can become extremely long, making them difficult to implement in industrial processes. In addition, with temperatures under 100 °C, the chemical composition of the oxide layers tends to be affected, and oxygen-rich films were obtained with a lower mass density and higher hydrogen content [54]. This phenomenon is reinforced by the findings of Verlaan et al. [55], showing a reduction in the hydrogen content of $AlO_x$ film from 13 at.% at 50 °C to under 1 at.% at 400 °C, while the mass density increases from 2.6 to 3.1 g cm$^{-3}$ in the same temperature range. This modified composition can significantly impact the oxidation characteristics and overall chemical stability, potentially leading to hydrolysis or other forms of degradation over time. Moreover, although Ylivaara et al. [56] noted no significant impact of deposition temperature on $AlO_x$ adhesion performance between 110 °C and 300 °C, it is plausible that reducing the deposition temperature below 100 °C could yield weaker adhesion and reduced interface quality. Additionally, the lower energy available at lower temperatures could impact the energetic processes that typically facilitate robust bond formation. These alterations in uniformity and chemical composition influence the barrier properties of the film. Supported via WVTR measurements of $AlO_x$ single layers deposited at 100 °C and 40 °C, a decrease in hermeticity by a factor of about 10 was demonstrated during this study.

## 4. Conclusions

This study systematically investigated the barrier properties of well-established barrier layers: silicon-based and metal oxide layers deposited via PECVD and ALD at low temperatures ($\leq$100 °C). To improve the barrier performance, fully inorganic and organic-inorganic combinations were evaluated. PECVD multilayers reached excellent WVTR values in the low $10^{-4}$ g m$^{-2}$ day$^{-1}$ range; however, this technique presents some weaknesses in terms of allowing the effective encapsulation of complex 3D components or systems. To overcome those limitations, parylene-$AlO_x$ solutions were combined to build up an organic–inorganic stack, demonstrating excellent conformality and resulting in WVTR values of close to $10^{-4}$ g m$^{-2}$ day$^{-1}$, with only one inorganic dyad. In conclusion, the combination of ALD metal oxide layers with parylene showed great promise in terms of achieving superior barrier properties and improved conformality while proving to be highly compatible with commonly used non-ideal substrate materials. This approach is particularly effective for high-performance 3D encapsulation applications, making it an ideal solution for use in such cases.

**Author Contributions:** Conceptualization, S.B., F.B., A.H. and J.B.; methodology, S.B., F.B., J.J.D.L. and A.H.; software, S.B.; validation, S.B., F.B., J.J.D.L. and A.H.; formal analysis, S.B.; investigation, S.B., F.B. and J.J.D.L.; resources, A.H. and J.B.; data curation, S.B.; writing—original draft preparation, S.B., F.B. and J.J.D.L.; writing—review and editing, S.B., F.B., J.J.D.L., A.H. and J.B.; visualization, S.B.; supervision, S.B., A.H. and J.B.; project administration, S.B., A.H. and J.B.; funding acquisition, S.B., A.H. and J.B. All authors have read and agreed to the published version of the manuscript.

**Funding:** This research was funded by Innosuisse, the Swiss Innovation Agency, via grant number 41363.1 IP-LS.

**Institutional Review Board Statement:** Not applicable.

**Informed Consent Statement:** Not applicable.

**Data Availability Statement:** The data presented in this study are available on request from the corresponding author.

**Acknowledgments:** The authors acknowledge the support of Ulrich Kroll and Yanik Tardy from Coat-X SA, who provided fruitful discussions and valuable insights. We also thank Olaf Zywitzki and his team at Fraunhofer Institute FEP, Dresden, for carrying out part of the water vapor transmission rate measurements (WVTR).

**Conflicts of Interest:** The authors declare no conflict of interest. The funders had no role in the design of the study; the collection, analyses, or interpretation of data; the writing of the manuscript, or the decision to publish the results.

## Appendix A

Figure A1 shows micro-channel structures on silicon wafers fabricated using a standard lithography process, used to assess coating conformality. Four different channel widths, namely 50, 100, 200 and 400 µm, all measuring 50 µm in depth, were manufactured on four-inch p-type silicon wafers (100). Prior to the deposition process, a covering glass slide was placed on the wafer to surround the channels, while an open cavity at the end of the channel ensured channel entry. In this study, only the coating penetration lengths in the 50 µm deep and wide channels were reported in order to calculate the AR.

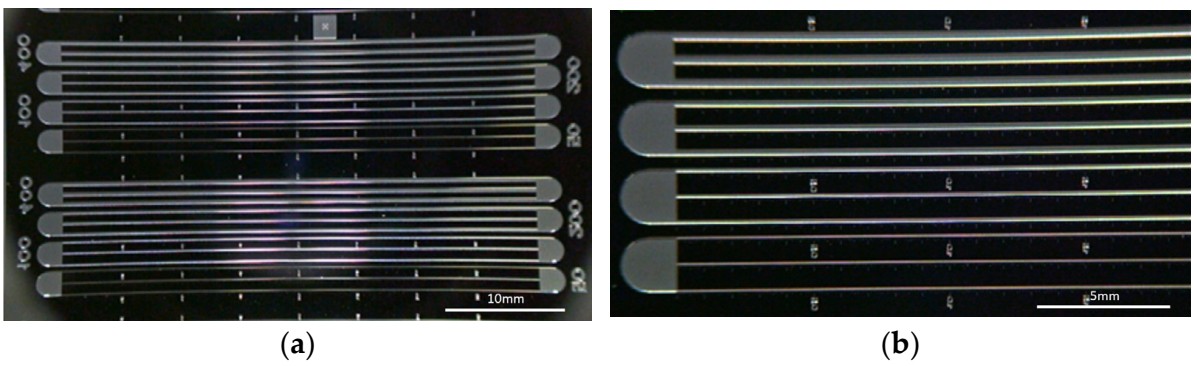

(**a**)　　　　　　　　　　　　　　　　　　　　　　　　　(**b**)

**Figure A1.** Silicon wafer micro-channel structures before deposition: (**a**) overall view of the four channel types; (**b**) partial view of the channel group showing the different widths (50, 100, 200 and 400 µm).

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
