# Peer review of "Parylene-AlOx Stacks for Improved 3D Encapsulation Solutions"

_coatings, doi:10.3390/coatings13111942_

Round 1

Reviewer 1 Report

Comments and Suggestions for Authors

The manuscript "Parylene-AlOx Stacks for Improved 3D Encapsulation Solu- 2 tions" by Buchwalder and coworkers describes the coating capabilities of some inorganic (SiO2, Al) and organic (parylene) materials with respect to wwater vapour permeability. The work is quite well written and may be interesting for Coatings' audience. Therefore, I recommend publication after fixing some few minor issues.

1) Please clarify the meaning (or give references) 2of some concentration data, such as PCS, HN, C or VT4 grade. They are not the easier technical, analytical or technical grade

2) Can you also provide photographs of your silicon wafers (maybe as additional material? It would  make it easier to visualize your device

3) Table 1 has one repeated row (first and last one are identical)

A couple of typos

was set to -> was set atlead to decrease mobility -> lead to decreased mobility (of the precursors)

Comments on the Quality of English Language

English is fine

Author Response

Dear Reviewer, 
Please see the attachment. Please don't hesitate to contact me if you have any questions or require further clarification.
Thank you very much,
Sébastien Buchwalder

Reviewer 2 Report

Comments and Suggestions for Authors

Manuscript ID: coatings-2702421

Authors reported an article entitled “Parylene-AlOx Stacks for Improved 3D Encapsulation Solutions”. Here, the authors finally concluded that the combination of ALD metal oxide layer with organic parylene should be promising in terms of barrier properties, conformality and others.  However, I suggest the authors may address the below issues for enhancing science as well as for reader-friendly.

1.      Authors may draw the chemical structures of organic polymers such as parylene, PET, PI, etc. Then address Tg and Tm of each polymer. Then if possible, please address the property change depending on temperature. For example, when T > Tg, there are much more fee volumes in polymer film, which makes WVTR be reduced. At least, authors should mention some physical properties of each polymer. Furthermore, the molecular weight information is very important in polymer science. In the case of metal oxide, if authors mention the crystal structure and properties, it will be helpful for reader-friendly.

2.      Line 127: “The pyrolysis temperature was set at 700 °C to cleave the dimers into monomers”

Question: Usually organic materials decompose at ~350-400 °C, but the authors’ processing temperature was 700 oC. How organic material could be survived at this high temperature? Authors may address some comments about it in the manuscript.

3.      Authors did not follow the abbreviation rule in several places. Some of them are as follows:

Line 81: polyethylene terephthalate => polyethylene terephthalate (PET)

Line 81: polyimide  => polyimide (PI)

Then Line 84 (PET) and Line 91 (PI), the abbreviation is enough.

Line 250: Figure 4’s Caption. Authors did not introduce “Px” which should be parylene. This abbreviation should be introduced here (Not Line 299) because it appears here for the first time.

4.      Line 130: “a low dielectric constant”   (Comment: Please input a typical value as an example)

5.      Equation (3) and Lines 189-191: Physical parameter should be italic. (Check all the places)

Author Response

(The authors gave the same response as above.)

Reviewer 3 Report

Comments and Suggestions for Authors

In this work, the author proposed a method of well-established silicon based PECVD and metal oxides ALD barrier coatings, which can be used in high-level 3D encapsulation applications. The subject considered is an interesting one and also has certain advantages in IC and MEMS field, which is fits the journal's scope. However, several concerns in the current stage of the manuscript need to be addressed before consideration of acceptance, as follows:

Q1. As shown in Figure 3, different units are used to measure the thickness of the film, should be described in uniform units.

Q2. By combining organic-inorganic layers greatly improves the barrier performances compared to single layer. However, it also reduces the flexibility of the device, so it is necessary to find a balance point to comprehensively improve the performance of the packaging process encapsulation performance.

Q3. In the part of barrier properties, merely structural and material studies are obviously inadequate. Investigations from the perspectives of operating principle are indispensable.

Q4. It is a typo in title of chapter 3.1.

Comments on the Quality of English Language

it is ok

Author Response

(The authors gave the same response as above.)
